# The Effects of a Glucocorticoid Receptor Agonist (GRA) on the Immune Function, Nutrient Digestibility, and Wean-to-Finish Growth Performance of Early-Weaned Pigs

**DOI:** 10.3390/ani10060953

**Published:** 2020-05-30

**Authors:** Hailey Wooten, Hwanhee Kim, Amanda R. Rakhshandeh, Anoosh Rakhshandeh

**Affiliations:** 1Department of Animal and Food Sciences, Texas Tech University, Lubbock, TX 79409, USA; hailey.wooten@ttu.edu (H.W.); hwanhee.kim@ttu.edu (H.K.); 2Department of Biology, South Plains College, Levelland, TX 79336, USA; arakhshandeh@southplainscollege.edu

**Keywords:** pig, glucocorticoid receptor agonist, antibiotics, inflammation, digestibility, growth

## Abstract

**Simple Summary:**

Early weaning is a common practice in modern swine production. Despite advantages, it exposes piglets to numerous stressors at an age when their digestive system is not fully developed. This exposure results in the hyper-activation of the immune system, which in turn reduces growth performance immediately following weaning. One common practice in the pig industry has been to use in-feed antibiotics to mitigate some of the negative effects of early weaning on the immune system and growth. However, recent concerns over antibiotic resistance have created a need for an alternative strategy. We previously have shown that a glucocorticoid-like compound can effectively improve growth performance of newly weaned pigs by regulating their immune response. In the present study, we evaluated the viability of this treatment as an alternative to in-feed antibiotics by directly comparing the effects of the two treatments on measures of immune function, nutrient digestibility, short-term growth and long-term growth. We found that treatment with glucocorticoid-like compounds reduces inflammation, improves nutrient digestibility and enhances short term growth. These effects in turn lead to long-term body weight superiority that is comparable to antibiotic treatment. In conclusion, treating early weaned pigs with glucocorticoid-like compounds is a suitable alternative to the use of in-feed antibiotics.

**Abstract:**

This study assessed the viability of glucocorticoid receptor agonist (GRA) treatment as an alternative to in-feed antibiotics (ANT) in wean-to-finish pigs. A total of 209 piglets were assigned to eight treatments based on a factorial arrangement, with GRA (+ vs. −; dexamethasone, 0.2 mg/kg body weight, BW), ANT (+ vs. −; 110 mg/kg in-feed Tylosin) and sex (gilt vs. barrow) as the main factors. The serial slaughter technique and serial blood collection were performed on 115 pigs during the first week post-weaning to collect blood, tissue and ileal digesta samples. Fecal samples were collected to determine energy digestibility. In comparison to ANT, GRA more effectively improved the measures of systemic inflammation, protein utilization and recovery-associated biomarkers (*p* ≤ 0.05). Relative to the control group, GRA treatment improved (*p* ≤ 0.03) dietary nutrient digestibility relative to control pigs, which was comparable to ANT effects. Relative to the control group, all groups had a higher ADG and BW during the starter phase (*p* < 0.01). Similar to the ANT group, GRA improved the gain-to-feed ratio relative to the control group during the starter phase. Relative to control pigs, overall BW was higher in GRA and ANT pigs during the grow-to-finish phase (*p* < 0.01). Collectively, these results suggest that GRA injection improves the growth performance of newly weaned pigs by reducing weaning-induced inflammation and improving nutrient digestibility. GRA can be used as an alternative to in-feed ANT to mitigate the effects of weaning stress on pigs.

## 1. Introduction

Early weaning in pigs is accompanied by an elevated level of intestinal and systemic inflammation that negatively impacts gut integrity, digestive capacity, and the overall health of the animal [1,2,3]. The culmination of these effects is a 5–10 day drop in growth commonly known as the post-weaning growth lag (PWGL) [4]. This period of growth is of particular importance because it can negatively affect the lifelong productivity of pigs [5,6,7]. Furthermore, evidence suggests that inflammation and gut dysfunction during early weaning results in lifelong hyper-reactivity to stress and immune challenges [8,9]. Recently, antibiotics have been fed during early weaning to promote gut health and reduce inflammation [10,11]. While antibiotics do positively influence the productivity of pigs, safety concerns now exist over the possible creation of antibiotic-resistant bacteria [12,13]. Therefore, alternative strategies for improving post-weaning productivity in pigs are needed. One such alternative, glucocorticoid receptor agonists (GRA), has been shown to improve growth during the period of time immediately following weaning [14]. Moreover, repeated injection of GRA, at a specific dose, lowers some measures of the inflammatory response and potentially reduces the redirection of nutrients toward the immune system [3]. However, to our knowledge the effects of GRA treatment have not been directly compared to antibiotic use. Whether the improved growth of GRA-treated pigs during the early post-weaning period will be maintained throughout the productive life of the pig is also unknown. Thus, the purpose of this study was to evaluate the impact of GRA treatment on selected measures of immune function, digestive capacity and nutrient digestibility in wean-to-finish pigs, as compared to antibiotics.

## 2. Materials and Methods

The experimental protocol for this study was reviewed and approved by the Animal Use and Care Committee of Texas Tech University (ACUC approval number 17022-02). All experiments were conducted at the Texas Tech University Swine Research center (New Deal, TX, USA). Dexamethasone (Phoenix Pharm Inc., Catalogue number PHS5731924305) was used as the GRA. Tylosin (Tylan 40, Elanco Inc., Greenfield, IN, USA) was used as the in-feed antibiotic.

### 2.1. Animals and Treatments

A total of 209 piglets (body weight, BW, 7.64, SD 1.33 kg; Pig Improvement Company, Hendersonville, TN, USA) were weaned at 26.0 ± 1.7 days of age, blocked by time (96 pigs in block 1 and 113 pigs in block 2), and randomly assigned to eight treatment groups based on a 2 × 2 × 2 factorial arrangement with GRA (+ vs. −), ANT (+ vs. −) and sex (gilt vs. barrow) as the main factors. Pigs in the GRA+ group received two intramuscular (i.m.) injections of GRA at −1 and 72 h post-weaning (0.2 mg/kg BW), as described previously by Wooten et al., 2019 [3]. Pigs in the ANT+ group received in-feed tylosin (110 mg/kg diet) during the first 7 days post-weaning. Pigs in the control group received i.m. sterile saline (0.5 mL/pig) to account for the stress of injection [14].

After weaning, piglets were moved into a temperature-controlled nursery and housed in floor pens (5–8 pigs per pen, 4 pens per treatment). On days 1, 3 and 5 post-weaning a total of 115 pigs were euthanized leaving 2–3 pigs per pen and 4 pens per treatment for the duration of the study (*n* = 94). Pigs were fed conventional corn-soybean meal (SBM) based diets throughout the study according to a phase feeding program (Table 1). Titanium dioxide (TiO_2_) was added to the diets as an indigestible marker during the starter phase. The inclusion rates were 0.4%, 0.3% and 0.25% in phase 1 (day 0–7), 2 (day 7–21) and 3 (day 21–28) starter diets, respectively. Body weight and feed intake (FI) were measured daily from days 1 to 7 and weekly until day 28 post-weaning. After 28 days post-weaning (i.e., the starter phase), pigs were transferred to a temperature-controlled grow to finish facility and BW and FI were measured every 3 weeks until 126 days post-weaning.

### 2.2. Sample Collection

Serial blood samples were collected every other day from a random subset of pigs in each treatment pen throughout the first week post-weaning (beginning at day 0 post-weaning) via the external jugular vein. Blood samples were collected in vacutainers (BD Company, Franklin Lakes, NJ, USA) containing sodium-heparin, ethylenediaminetetraacetic acid (EDTA), or lithium-heparin. Samples collected in sodium-heparin tubes were centrifuged at 1500× *g* for 10 min at 4 ℃ and the plasma fraction was collected and stored at −80 ℃ until further analysis. Samples collected in EDTA vacutainers were sent on ice to the Texas A & M Veterinary Diagnostic Laboratory (Amarillo, TX, USA) and analyzed within 24 h of collection for complete blood cell counts (CBC), total plasma proteins, and fibrinogen.

At 1, 3 and 5 days post-weaning 39, 38 and 38 pigs were euthanized, respectively (1–2 pigs/pen), as follows: pigs were first sedated with an i.m. injection of a TKX (Telazol reconstituted with 2.5 mL 100 mg/mL ketamine, and 100 mg/mL 2.5 mL xylazine) mixture at 0.03 mL TKX/kg BW, and then euthanized by lethal injection of sodium-pentobarbitol (Fatal-Plus; 1 mL/10 lbs BW) to lessen stress and intestinal mucosal shedding during euthanasia [15]. Immediately after euthanasia, a mid-ventral laparotomy was performed on each pig to harvest jejunum and digesta samples. About 40 cm of the jejunum was taken at approximately 1 m distal to the gastroduodenal junction and divided into three segments. One undamaged segment (~4 cm) was stored in 10% formalin and used for histomorphology analysis of the intestinal lining. The remainder of the jejunum sample was rinsed with phosphate buffered saline (PBS) and a second segment was used for collecting mucosal scrapings, by flash freezing adherent material in liquid nitrogen and storing it at −80 ℃ [3,16]. The remaining third segment of jejunum was flash frozen and stored at −80 ℃ until further analysis. Digesta was collected from the distal portion of the ileum, as described previously [17], immediately flash-frozen in liquid nitrogen, and stored at −20 ℃. Fresh fecal samples were collected from each treatment pen at weekly intervals until the end of the starter phase (i.e., days 7, 14, 21 and 28 post-weaning) and stored at −20 ℃ until pooled and homogenized for each pen and collection period. Fecal homogenates and ileal digesta samples were lyophilized (Freezone 4.5, labconco, Kansas City, MO, USA) prior to proximate and TiO_2_ analysis.

### 2.3. Sample Processing and Analysis

The Comparative Pathology Core at Iowa State University (Ames, IA, USA) evaluated morphological measurements of the jejunum mucosa. Using the hematoxylin and eosin staining method, each sample had two villi and two crypts from five separate fields (a total of 10 villi and 10 crypts per sample) evaluated to establish villus height and crypt depth. Villus density was measured by dividing the number of villi within each of the five fields by the length (mm) of each field. Complete blood cell count (CBC) analysis was conducted using a ProCyte Dx (IDEXX Laboratories, Inc., Westbrook, ME, USA) and microscope differential to provide both instrument data and microscopic evaluation of samples at the Texas A&M Veterinary Medical Diagnostic laboratory (Amarillo, TX, USA). Blood chemistry was evaluated using handheld iSTAT (Abbott Point of Care Inc., Princeton, NJ, USA) analysis within 30 min of blood collection, as described previously by McGilvray et al. [17].

The level of alkaline phosphatase (AP) in the jejunum was measured in homogenized frozen jejunum tissue using a tissue protein extraction reagent (Thermo Scientific, Rockford, IL, USA) and a tissue homogenizer (Homogenizer 850, Fisher Scientific, Pittsburgh, PA, USA). Samples were then centrifuged at 1000× *g* for 16 min at 4 ℃. After centrifugation, the supernatant was collected and the AP concentration was measured using a Porcine AP ELISA kit (MyBioSource Inc., San Diego, CA, USA) according to the manufacturer’s instructions. Jejunum levels of heat-shock protein 70 (HSP-70) and aminopeptidase N (APEP) were measured in homogenized frozen jejunum tissue in PBS with a tissue homogenizer (Homogenizer p850). After homogenization, mixtures were subjected to two freeze–thaw cycles to break the cell membranes and centrifuged at 2000× *g* for 12 min before collecting the supernatant. The concentration of HSP-70 was measured in the supernatant using a Pig Heat Shock Protein 70 (HSP-70) ELISA kit (Mybiosource Inc., San Diego, CA, USA) according to the manufacturer’s instructions. The concentration of aminopeptidase N was measured in the supernatant using a pig aminopeptidase nitrogen (N) sandwich ELISA kit (Mybiosource Inc., San Diego, CA, USA) according to the manufacturer’s instructions. Jejunum levels of sucrase isomaltase (SI) and maltase glucoamylase (MGA) were measured by homogenizing frozen jejunum tissue in PBS with a tissue homogenizer (Homogenizer 850), followed by centrifugation at 1000× *g* for 20 min before collecting the supernatant. The sucrase isomaltase concentration was measured in the supernatant with a Porcine Sucrase Isomaltase ELISA kit (Mybiosource Inc., San Diego, CA, USA) according to the manufacturer’s instructions. The maltase glucoamylase concentration was measured in the supernatant with a Porcine Maltase Glucoamylase, Intestinal (MGA) ELISA kit (Mybiosource Inc., San Diego, CA, USA) according to the manufacturer’s instructions. Total protein concentration for homogenized tissue extracts was measured using a Pierce bicinchoninic acid (BCA) protein assay (Thermo Fisher, Rockford, IL, USA) following the manufacturer’s procedures. Jejunum enzyme and heat-shock protein abundance were expressed relative to the total protein concentration in the homogenized tissue extract.

Apparent ileal digestibility (AID) of crude protein (CP) and apparent total tract digestibility (ATTD) of gross energy (GE) were measured in the digesta and pooled fecal samples, respectively, using an indicator method [17]. The levels of CP in the samples were evaluated using the combustion method (LECO Corporation, St. Joseph, MI, USA), according to McGilvray et al. [17]. The gross energy of samples was determined using bomb calorimetry at the Washington State University Wildlife Habitat and Nutrition Laboratory (Pullman, WA, USA). Levels of TiO_2_ in fecal, digesta and diet samples were evaluated using the Association of Official Analytical Chemists (AOAC) calorimetric method [18].

### 2.4. Statistical Analysis

Statistical analyses were performed using the MIXED procedures (PROC MIXED) in SAS (version 9.4, SAS Institute, Cary, NC, USA). Normality and homogeneity of variances were confirmed using the univariate procedure (PROC UNIVARIATE). Outliers were determined as any value that differed from the treatment mean by ± two standard deviations. Data were analyzed using a factorial randomized complete block design, with block, GRA, ANT, SEX, day and interaction among the main factors (GRA × ANT × SEX × day × block) as fixed effects. For performance parameters and ATTD of GE, pen and feed intake were used as the experimental unit and co-variate, respectively. For AID of CP, pig and feed intake were used as the experimental unit and co-variate, respectively. For growth performance analysis, final BW of each phase (i.e., BW at day 0 and 28 post-weaning) was used as co-variate, and when appropriate (*p* > 0.10) the reduced model was used. For all other analyses, pig was used as the experimental unit. Pig within pen was used as a random effect. For all parameters that were measured repeatedly over time, repeated measurements analysis of variance was used. An autoregressive covariance structure was used for repeated measurements. Values were reported as least squares means ± largest SE. Treatment means were separated using the Tukey test. Significance was considered at *p* ≤ 0.05 and a tendency was considered at *p* ≤ 0.10.

## 3. Results

During the first three weeks post-weaning, pigs did not contract or show any clinical symptoms of major swine diseases. However, one pig in block 1 was removed from the study near the end of the starter phase due to infection with *Actinobacillus suis*. Two other pigs were also removed from the study at days 40 and 25 post-weaning in blocks 1 and 2, respectively, due to lameness. Statistical analysis revealed no outliers; therefore, no other measurements were removed from the study. The rest of the pigs were healthy and did not show any clinical symptoms of disease during the course of the study. Analyzed nutrient contents of the diets were within 5% of the calculated values.

### 3.1. Complete Blood Cell Count and Blood Chemistry

Complete blood cell count (CBC) results are presented in Table 2. Data on the effects of treatments on CBC over time are presented in Appendix A. No significant three-way interaction effects between the main factors (i.e., GRA × ANT × SEX) were observed for CBCs. An interaction effect between the main factors and day (GRA × ANTf × SEX × day) was only observed for plasma fibrinogen concentrations (Appendix A; *p* = 0.02). Pigs treated with GRA had lower total white blood cell counts (WBC) than control pigs [17.5 vs. 14.7 ± 0.73 thousands per microliter (K/μL) of blood; *p* = 0.005]. Compared to the control treatment, GRA and ANT treatments, but not a combination of the two, reduced lymphocyte counts (*p* = 0.03). Treatments had no effect on the neutrophil count (*p* > 0.50). The neutrophil-to-lymphocyte (N:L) ratio was also not affected by the treatments, but a GRA by day interaction resulted in a lower N:L ratio compared to the control group during the first week post-weaning (Appendix A; *p* = 0.03). Relative to the control treatment, the eosinophil count was lowered by both GRA and ANT treatments. However, GRA treatment reduced the eosinophil count to a greater extent than ANT treatment during the first week post-weaning (*p* = 0.01). Treatment with GRA, but not ANT, increased the platelet count (857.7 vs. 727.8 ± 47.37 K/μL; *p* = 0.05). Compared with the control group, GRA or ANT treatments reduced the total plasma protein concentration, with GRA treatment reducing the concentration to a higher extent (*p* = 0.01), predominantly in gilts (*p* = 0.03), than did ANT treatment. Treatment with GRA or ANT reduced the concentrations of plasma fibrinogen compared with the control treatment. However, GRA treatment reduced the plasma fibrinogen concentration to a higher extent than did ANT treatment (*p* = 0.03). Relative to the control group, the concentration of fibrinogen was effectively lowered by GRA in gilts on days 1, 2 and 7 post-weaning, but only on day 7 post-weaning in barrows (Appendix A; *p* < 0.05). In contrast to GRA treatment, ANT treatment was only effective in lowering fibrinogen levels on day 7 post-weaning in barrows (*p* < 0.05). No effects of treatments or day on red blood cell counts, neutrophil counts, hemoglobin levels, hematocrit, packed cell volume, mean corpuscular volume, mean corpuscular hemoglobin or mean corpuscular hemoglobin concentrations were observed (*p* > 0.50).

Blood chemistry data are presented in Table 2. Data on the effects of treatments on measures of blood chemistry over time are presented in Appendix A. There were no significant three-way interaction effects among GRA, ANT and SEX (GRA × ANT × SEX) on measured parameters (*p* > 0.50), with the exception of blood glucose concentrations. No interaction effects between the main factors and day (GRA × ANT × SEX × day) were observed for measures of blood chemistry (*p* ≥ 0.10). A tendency (*p* = 0.06) for a GRA × ANT × SEX effect on blood glucose concentrations was observed. While in all other treatment groups (i.e., GRA−ANT+, GRA+ANT+ and GRA−ANT−) the blood glucose concentrations were higher in barrows than gilts, in GRA-treated pigs, blood glucose was higher in gilts than barrows. Pigs treated with GRA had lower levels of blood urea nitrogen (BUN; 7.63 vs. 5.82 ± 0.613 mg/dL; *p* = 0.04) and creatinine (1.23 vs. 1.11 ± 0.041 mg/dL; *p* = 0.02) compared with control pigs. Other parameters, such as electrolyte balance, anion gap and hemoglobin, were not affected by the treatments (*p* ≥ 0.10). Blood urea nitrogen and creatinine were lower (*p* = 0.01) and tended to be lower (*p* = 0.08), respectively, in the pre-weaning (i.e., −1 day post-weaning) period than in the first week post-weaning. The values (pre-weaning vs. post-weaning) were 3.7 vs. 7.7 ± 0.82 and 1.08 vs. 1.20 ± 0.046 for BUN and creatinine, respectively. 

### 3.2. Histomorphology of the Intestinal Lining, Intestinal Enzymes, and Nutrient Digestibility

The effects of experimental treatments on villus height and density, as well as crypt depth, are presented in Table 3. No significant three-way interaction effects of GRA, ANT and SEX (GRA × ANT × SEX) on measures of histomorphology were observed (*p* > 0.10). Further, no interaction effects between the main factors and day (GRA × ANT × SEX × day) were observed for the measures of blood chemistry (Appendix A; *p* ≥ 0.10). The GRA by ANT interaction was significant for the crypt depth (*p* = 0.03). Relative to the control group (i.e., GRA−, ANT−), crypt depth was higher in pigs that received GRA without ANT (GRA+ANT−). Similarly, only pigs treated with GRA (GRA+ANT−) had a higher villus density relative to control pigs (9.72 vs. 8.88 ± 0.230 mm; *p* = 0.05). Villus height was decreased by GRA treatment (0.24 vs. 0.27 ± 0.006 mm; *p* < 0.01), but increased by ANT treatment (0.27 vs. 0.25 ± 0.006 mm; *p* = 0.03). The villus height-to-crypt-depth ratio was decreased by GRA treatment (1.05 vs. 1.18 ± 0.036; *p* < 0.01).

Data on the impact of experimental treatments on the relative concentrations of digestive enzymes and heat-shock protein 70 are presented in Table 3. Data on the effects of treatments on intestinal enzymes over time are presented in Appendix A. The relative concentration of SI was higher in the jejunum of GRA+ barrows compared to GRA+ gilts and GRA− barrows (*p* = 0.05), but was not different from GRA- gilts (*p* = 0.34). The relative concentration of jejunum SI was lower in the GRA−ANT+ and ANT+GRA+ groups than in the GRA−group (*p* < 0.05). The interaction among the main factors of GRA, sex and time (GRA × sex × day) was significant (Appendix A): when treated with GRA, the SI relative concentration was elevated in barrows on day 1 (67.4 vs. 43.3 ± 5.75 pg/mg protein; *p* < 0.01) but not on days 3 and 5 (*p* ≥ 0.10). Further, the interaction among the main factors of ANT, sex and day (ANT × sex × day) was significant: relative to their non-ANT treated counterparts, on day 1 post-weaning, the SI concentration was lower (*p* < 0.01) in ANT-treated gilts (76.1 vs. 43.7 ± 5.80) and barrows (70.3 vs. 40.4 ± 6.16), respectively. However, on day 3 post-weaning, ANT-treated gilts had higher jejunum SI levels than gilts in the ANT- group (151.485 vs. 129.7 ± 5.93 pg/mg of protein; *p* ≤ 0.01). The concentration of SI was significantly increased on days 3 and 5 post-weaning relative to day 1 (57.6, 150.0, and 124.1 ±4.36 ng/mg protein at days 1, 3 and 5 post-weaning, respectively; *p* < 0.01).

The interaction effects between the main factors and day (GRA × ANT × SEX × day) was significant for MGA (Appendix A; *p* < 0.04). Only on day 1 post-weaning was the concentration of MGA higher in gilts treated with a combination of GRA and ANT (GRA+ANT+), as compared to their barrow counterparts (8.64 vs. 7.83 ± 1.610 ng/mg protein; *p* < 0.01). The concentration of MGA tended to increase at day 5 post-weaning relative to days 1 and 3 (6.92, 6.21 and 8.52 ± 0.738 ng/mg protein at days 1, 3 and 5 post-weaning, respectively; *p* = 0.10).

For the relative concentration of APEP, no three-way interaction effects between GRA, ANT and sex (GRA × ANT × SEX) were observed. The interaction between the main factors of GRA and ANT was significant for APEP (*p* = 0.04): relative to the control and GRA groups, pigs treated with a combination of GRA and ANT (GRA+, ANT+) had higher concentrations (*p* = 0.05) of APEP than those treated with only ANT (GRA−, ANT+). Further, there was an interaction between the three main factors and time (GRA × ANT × SEX × day) for APEP concentrations (Appendix A; *p* < 0.01): treatments variably affected the relative concentrations of APEP over the first 5 days post-weaning. On day 1 post-weaning, GRA−ANT+ gilts and GRA+ANT+ barrows had the highest (*p* ≤ 0.05) APEP levels in the jejunum, while GRA−ANT+ barrows had higher APEP levels than control barrows. On day 3 post-weaning, gilts in GRA−treated groups had higher APEP levels relative to control gilts and barrows, but there were no differences between other treatment groups and control barrows (*p* ≥ 0.10). On day 5 post-weaning, GRA−ANT+ gilts and barrows had higher APEP concentrations than all other treated groups (*p* ≤ 0.05). Only GRA+ANT- gilts had lower APEP concentrations than control gilts and barrows on day 5 post-weaning (*p* = 0.03).

Relative to days 1 and 3 post-weaning, the concentration of AP was significantly reduced on day 5 in the jejunum of pigs (16.3, 17.8, and 12.9 ± 1.12 ng/mg protein at days 1, 3 and 5 post-weaning, respectively; *p* < 0.01). Treatment with GRA had no effect on the relative concentration of AP in the jejunum (*p* = 0.69). However, pigs in the ANT+ group had a lower level of AP in their jejunum than ANT− pigs (*p* < 0.01). Gilts had a lower level of AP than did barrows (*p* = 0.04). No GRA × ANT × SEX × day interaction effects were observed for AP (Appendix A; *p* = 0.13). However, two-way interactions between the main effects of GRA and day (GRA × day), as well as ANT and day (ANT × day), were observed: GRA treatment reduced the relative concentration of AP in the jejunum of gilts on day 1, and in the jejunum of both barrows and gilts on day 5 post-weaning (*p* < 0.01). In comparison, ANT treatment reduced the relative concentration of AP in both gilts and barrows on days 1 and 5 post-weaning (*p* < 0.01). Administration of a combination of GRA and ANT had no effect on AP concentration in the jejunum of pigs (*p* = 0.34). Relative to days 1 and 3 post-weaning, the concentration of AP was significantly reduced on day 5 in the jejunum of pigs (16.3, 17.8 and 12.9 ± 1.12 ng/mg protein at days 1, 3 and 5 post-weaning, respectively; *p* < 0.01).

No significant three-way interaction effects of GRA, ANT and SEX on the relative concentrations of jejunum enzymes were observed (*p* ≥ 0.10), with the exceptions of HSP-70 and SI (*p* ≤ 0.01). The interaction effects between the main factors and day (GRA × ANT × SEX × day) were significant only for the relative concentration of HSP-70 (*p* < 0.01). Relative to other treatment groups, GRA increased HSP-70 concentrations in the jejunum of both gilts and barrows (*p* < 0.02). In comparison, ANT treatment reduced HSP-70 levels in the jejunum of barrows, compared to the control group, without affecting the relative concentration in gilts (*p* < 0.02). Treatment with a combination of GRA and ANT (GRA+ANT) reduced jejunum HSP-70 to undetectable levels in both gilts and barrows. Treatment with GRA increased the relative concentration of HSP-70 in the jejunum of both gilts and barrows on day 5 post-weaning, compared with the control group (Appendix A; *p* < 0.01). The relative concentrations of HSP-70 were increased to a statistically significant level only at day 5 post-weaning (5.14, 0.20 and 123.03 ± 10.689 ng/mg protein at days 1, 3 and 5 post-weaning, respectively; *p* < 0.01).

Digestibility results are presented in Table 3. Data on the effects of treatments on nutrient digestibility over time are presented in Appendix A. No three-way interaction effects between the main factors (GRA × ANT × SEX) were observed for ATTD of dietary GE or AID of dietary CP. All pigs that received the GRA treatment had a higher ATTD of GE than those not treated with GRA (0.633 vs. 0.571; *p* < 0.01) during the 28-day post-weaning period (i.e., the starter phase). The interaction effect between GRA and ANT on AID of dietary CP was significant (*p* =0.01): treatment with GRA or ANT, but not the combination of both, increased AID of CP during the first 5 days post-weaning (*p* ≤ 0.04). The interaction between the main factors and time (GRA × ANT × SEX × day) was significant for AID of CP (*p* = 0.02): unlike other treated groups, pigs treated with only GRA (GRA+ANT−) had a consistently higher AID of CP relative to control pigs on days 1, 3 and 5.

### 3.3. Growth Performance

#### 3.3.1. Nursery Period (0 to 28 Days Post-Weaning)

Growth performance results for the nursery period are presented in Table 4. No three-way interactions (GRA × ANT × SEX) between the main effects and the measured parameters of growth performance were observed during the starter phase (*p* ≥ 0.30). A two-way GRA × ANT interaction was observed for ADG (*p* < 0.02): relative to the control group, all GRA- and ANT−treated groups had a higher ADG during the nursery period, but the GRA-treated pigs (GRA+ANT−) had the highest ADG (*p* ≤ 0.01). The main effect of SEX was also significant for ADG, with gilts outperforming barrows (0.34 vs. 0.31 ± 0.013 kg/d; *p* = 0.05). Similarly, there was a GRA by ANT (GRA × ANT) interaction for BW, as all groups had a higher BW relative to the control group during the starter phase (*p* ≤ 0.01). Relative to the control group, GRA−ANT+, GRA+ANT− and GRA+ANT+ pigs had a higher gain-to-feed (G:F) ratio, but the GRA+ANT− treatment increased the G:F ratio to a larger extent than the GRA−ANT+ and GRA+ANT+ treatments (GRA × ANT interaction; *p* < 0.01). An interaction between the main factors of GRA, ANT and time (GRA × ANT × day) for the G:F ratio was also observed: the higher G:F ratio in GRA+ANT−, GRA−ANT+ and GRA+ANT+ pigs was only observed in the first two weeks of the starter phase (*p* < 0.01). No treatment effects on the average daily feed intake (ADFI) were observed throughout the nursery period (*p* ≥ 0.20).

#### 3.3.2. Grow-to-Finish Phase (49 to 126 Days Post-Weaning)

Data are presented in Table 5. No three-way interactions between the main factors (GRA × ANT × SEX) and the measured parameters were observed (*p* ≥ 0.20). While GRA and ANT treatment did not affect the estimate of ADG in the grow-to-finish phase, an effect of SEX was observed, with barrows outperforming gilts (0.91 vs. 0.86 kg/d; *p* = 0.04). The mean BW for the grow-to-finish period was higher in GRA− (63.7 ± 2.76 kg) and ANT− (63.43 ± 2.76 kg) treated pigs than in control (60.0 ± 2.73 kg) pigs (*p* ≤ 0.045). Treatments had no significant effect on the G:F ratio (*p* ≥ 0.20). The main effect of sex was significant for ADFI, with barrows having a higher ADFI than gilts (2.50 vs. 2.36 kg/d; *p* < 0.01).

## 4. Discussion

During early weaning, pigs experience elevated levels of stress, referred to as the stress of weaning, which often leads to intestinal and systemic inflammation, and consequently gut dysfunction and repartitioning of nutrients away from growth [17]. As a result, a severe reduction in growth during the first seven to ten days post-weaning is observed [2,19,20]. While detrimental during the weaning period, evidence suggests that the physiological effects of weaning also lead to lifelong impairments in gut health, as well as sensitivity to stress and immune challenges [8,9,21,22]. Moreover, the period of growth immediately following weaning directly impacts how fast pigs reach market weight [5,6]. Historically, antibiotics have been used as a means of improving gut health and reducing inflammation during weaning. However, an alternative to antibiotic use is needed, as antibiotics pose safety risks in the form of antibiotic resistance [12,13]. We have previously shown that GRA treatment, if applied in the right dose, decreases measures of inflammation in pigs, both locally and systemically, and improves growth performance during the immediate post-weaning period and throughout the starter phase [3]. To further investigate the viability of GRA treatment as an alternative to in-feed antibiotics, we directly compared the effects of both on physiological markers of gut and immune health during weaning. Additionally, we explored whether the post-weaning benefits, conferred by GRA treatment, would lead to improvements in growth performance during the growing and finishing phases, as compared to the in-feed application of antibiotics. In this study, dexamethasone was chosen as the GRA because its anti-inflammatory properties are well studied, it is frequently utilized in the livestock industry, its short half-life eliminates the possibility of any residues at slaughter, and the short half-life minimizes the possibility of changes in body composition at the dose used in this study [3,23]. Finally, we evaluated nutrient digestibility during the nursery period (0 to 28 days post-weaning) to further explore the underlying mechanisms through which GRA treatment positively affects the growth performance of piglets, compared to antibiotic use.

In the present study, overall decreases in WBC, lymphocyte and eosinophils counts, as well as total plasma proteins and the acute-phase protein (APP), fibrinogen, suggest that treatment with GRA effectively mitigated the weaning-induced systemic inflammatory response in pigs [24]. These anti-inflammatory effects of GRA are in agreement with the results of our previous studies, in which we observed that repeated injection of GRA suppressed measures of the inflammatory response, such as the levels of APPs and pro-inflammatory cytokines [3]. Compared with GRA, ANT treatment had a more moderate and slower effect on measures of immune function, which is consistent with the findings of other workers [10,11]. In the present study, GRA treatment increased the N:L on day 3 post-weaning. A high N:L ratio is often associated with higher levels of inflammation, which could suggest that GRA−treated pigs were experiencing more inflammation [25]. However, recent studies have shown that glucocorticoids have pleotropic effect on leukocytes. Specifically, glucocorticoids suppress the proliferation and differentiation of most leukocytes, including lymphocytes, while having no effect on neutrophils. An inflammatory response, on the other hand, typically shows a greater increased proliferation of neutrophils, relative to lymphocytes, which increases the N:L ratio. Antibiotic treatment affects the N:L ratio in yet a different way, specifically by suppressing neutrophil proliferation to a greater extent than lymphocyte proliferation [25,26]. Therefore, in the present study, the increase in the N:L ratio in GRA−treated pigs may be due to a lack of GRA influence on neutrophils, while at the same time decreasing lymphocyte numbers, rather than an increase in actual inflammation. This idea is further supported by the reduced levels of creatinine and BUN, measures of inflammation, that were observed in GRA−treated pigs in the present study. This result may also explain why in this study and other studies, weaning itself, which is accompanied by an increase in endogenously produced glucocorticoids (cortisol), does not influence the N:L ratio [24].

In the current study, treatment with GRA, but not ANT, substantially increased the platelet count in pigs, suggesting that GRA−treated pigs likely had a faster recovery from the inflammatory state than pigs receiving other treatments [27]. This idea is further supported by the enhanced concentration of HSP-70 in GRA−treated pigs. Heat shock protein 70 is a molecular chaperone that reduces inflammation and helps maintain intestinal integrity [28,29]. Pigs exposed to the stress of weaning and transport show lower HSP-70 concentrations in the small intestine [30]. The latter agrees with the findings of the current study, in which only GRA-treated barrows and gilts could maintain their HSP-70 concentrations at a detectible level in the jejunum. Thus, the increase of HSP-70 in GRA−treated pigs on day five may be associated with lower levels of intestinal inflammation and faster intestinal recovery from the damage caused by weaning.

In comparison to GRA or saline (control) treatment, ANT treatment reduced the concentration of AP in the jejunum of pigs during the first five days post-weaning in the present study. The reduced AP concentration in ANT−treated pigs in our study agrees with the findings of other workers, who reported that antibiotics characteristically cause dysbiosis in the number and composition of intestinal commensal microbiota, as well as reduced concentrations of intestinal AP [31,32]. Reduced AP concentrations were more evident in ANT−treated pigs, particularly at day five post-weaning, most likely in response to a more prolonged exposure to ANT. In contrast, glucocorticoids have been shown to upregulate intestinal AP activity and mRNA expression in canines and other species [31,33]. This finding is often associated with increased intestinal resistance to microbial endotoxins that are often produced by luminal microbiota [34]. However, in the current study, GRA treatment did not affect the overall mean concentration of AP during the first five days post-weaning, which may suggest that GRA treatment, at the dose that was used in the current study, was not large enough to enhance AP concentrations. On the other hand, this result could also suggest that GRA treatment may have had a minimal effect on microbiota diversity, since AP concentrations in the jejunum and microbiota diversity often change coordinately. Regardless, higher levels of AP in the jejunum suggest that the GRA-treated pigs may have a higher capacity to resist microbial endotoxins, which may, in part, explain why GRA-treated pigs had lower levels of inflammation compared to ANT- and saline-treated pigs.

In the current study, GRA treatment, but not ANT treatment, effectively reduced the BUN and blood creatinine in pigs. Increased blood creatinine (an indicator for increased skeletal muscle degradation) and BUN (the primary metabolite derivative of AA catabolism) are characteristics of systemic immune system stimulation and an inflammatory state [17,26,35]. Thus, reduced creatinine and BUN concentrations in the GRA−treated pigs in the present study further indicate, along with reduced levels of APPs, that these pigs had a lower level of systemic inflammation. The latter provides an explanation, in part, for why GRA-treated pigs had a higher G:F ratio than other treatment groups, particularly during the first two weeks post-weaning. Lower levels of systemic inflammation are associated with a higher availability of nutrients for growth [17]. The findings of the current study are consistent with the findings of Sterndale et al., who also reported that GRA treatment reduced BUN post-weaning [36]. Finally, the observed higher blood glucose concentrations in barrows as compared to gilts can likely be attributed to the higher level of endogenous glucocorticoids found in barrows, since feed intake was not affected by the treatments. Specifically, glucocorticoids increase gluconeogenesis in the liver of animals, thus causing higher blood glucose levels [37,38].

From a morphological standpoint, GRA treatment reduced intestinal villus height relative to the control treatment, with barrows being more affected than gilts. The latter can likely be attributed to the pleiotropic effects of glucocorticoids, which downregulate protein synthesis in the submucosal layer of the jejunum, especially in barrows [9,39]. Barrows have higher basal glucocorticoid concentrations than gilts, which likely explains, in part, why the villus height of GRA−treated barrows was more affected than their gilts counterparts in the current study [39]. This study also found that pigs treated with only GRA had a higher villus density relative to control pigs. While increased villus height has long been associated with more surface area for nutrient absorption [9], the findings of the current study suggest that villus density may be a better indicator of absorptive capacity, since GRA−treated pigs experienced higher ADG, BW and G:F ratio, despite shorter villi, than control and ANT−treated pigs [3]. However, the importance of villus height versus villus density requires further investigation, since gut morphology is not the only factor in the digestion and absorption of nutrients. In the present study we did not observe a relationship between the relative concentrations of key intestinal digestive enzymes (i.e., SI, MGA and APEP) and measures of nutrient digestibility and growth. The effects of GRA and ANT treatment on the relative concentrations of these enzymes were variable and were more influenced by time rather than the experimental treatments. Nutrient digestibility, on the other hand, is a reliable measure for nutrient bioavailability [40]. In the current study, repeated i.m. injection of GRA resulted in a higher dietary ATTD of GE and AID of CP in pigs, compared with the untreated group. Treatment with GRA likely improved these measures of digestibility by reducing inflammation-induced intestinal damage and mucus secretion [3,41,42]. The mucin fraction of mucus is the main component of intestinal endogenous protein losses, and directly affects the ileal digestibility of dietary protein, leading to a reduced availability of proteins and amino acids for growth [40]. In comparison with GRA treatment, ANT treatment did not influence ATTD of GE and only influenced AID of CP on day five, suggesting a slower intestinal healing process in ANT−treated pigs. This idea is supported by other findings in the current study that showed that GRA treatment more effectively increased physiological markers of healing, such as platelet counts and HSP-70 concentrations, compared with ANT treatment. Collectively, these results suggest that treatment with GRA can improve the AID of dietary CP and the ATTD of dietary GE, most likely by subsiding intestinal inflammation and reducing intestinal protein losses. Thus, the higher G:F ratio in GRA-treated pigs during the nursery period may, in part, be associated with the higher availability of energy and protein for this group of pigs.

In the present study, relative to the untreated control group, treatment with GRA and/or ANT improved ADG, feed efficiency and overall BW during the nursery period. During the starter phase, the effects of GRA treatment on measures of growth performance were generally either superior or comparable to ANT treatment. These findings are in agreement with previous studies, where GRA treatment improved measures of growth performance during the post-weaning period [3]. In addition, the current findings suggested that GRA and ANT treatment are most effective at improving active growth during the first two weeks post-weaning. This idea is supported by the finding that GRA and/or ANT treatment had no effect on ADG and the G:F ratio during the grow-to-finish period. However, despite this finding, BW remained higher in the GRA− and ANT-treated pigs throughout the study, providing evidence for the impact of superior growth performance during the starter phase, particularly in the first two weeks post-weaning, on the overall productivity of pigs [5,6]. One could argue that the higher ADG of GRA−treated pigs in the current study could be a result of enhanced water retention or increased lipid deposition, since increased whole-body water retention and lipid deposition has been associated with medium- to long-term use of high doses of synthetic glucocorticoids [43,44]. However, increased water retention likely does not explain the findings in our study, because (1) we used a very low dose of dexamethasone, which is three times lower than the immunosuppressive dose recommended for pigs [3], (2) we used short-term administration of dexamethasone as a GRA, (3) the short half-life of dexamethasone in pigs reduces the likelihood of increased, long-term water retention in GRA-treated pigs, and (4) the ADG and G:F ratio remained superior in GRA−treated pigs throughout the starter phase, which extended far beyond the half-life of dexamethasone. In addition, dexamethasone administration likely did not increase the lipid deposition of GRA−treated pigs in this study because the G:F ratio remained superior in GRA-treated pigs throughout the starter phase. It also remained unaffected during the growing and finishing phases in GRA−treated pigs, further suggesting that lipid deposition was not increased in these animals [40]. Hence, the superior growth performance of GRA−treated pigs is likely explained by better digestive function, due to lower levels of inflammation.

Using data from the present study, one can predict that GRA treatment and ANT treatment may potentially reduce the time needed to reach 120 kg BW by roughly 3.5 and 0.9 days, respectively. The latter is in agreement with other researchers, who reported that higher ADG during the immediate post-weaning period results in fewer days to reach market weight [5,6]. Finally, in the present study, an overall trend for ADG to be higher in gilts than in barrows was observed during the nursery period. This trend was reversed during the grow-to-finish period. These observations are in agreement with the findings of other workers, who reported that gilts perform better in response to weaning, while barrows often experience compensatory growth during later stages of life [45,46]. In the current study, the observed superior ADG of barrows in the grow-to-finish phase did not lead to improved feed efficiency in these animals, since the ADFI of barrows was significantly higher during the grow-to-finish phase.

## 5. Conclusions

The results presented here corroborate previous findings in which repeated i.m. administration of GRA, at the dose used in the current study, subsides intestinal and systemic inflammation in the first week post-weaning. Furthermore, the positive effect of GRA on growth performance may be associated with a reduction in muscle protein degradation, and improvements in overall intestinal health and apparent dietary nutrient digestibility. When comparing two mitigating strategies for the PWGL, GRA administration seemed to impact parameters associated with whole-body inflammation, and hence nutrient utilization, to a greater extent than ANT treatment. Additionally, GRA treatment effectively improved measures of growth performance and feed efficiency, which was comparable, or even superior in some aspects, to the effects of in-feed ANT. Furthermore, while ADG was not improved by GRA treatment during the grow-to-finish period, GRA pigs, similar to ANT pigs, maintained BW superiority until 126 days post-weaning, suggesting that GRA treatment may convey economic advantages by reducing the time needed to reach market weight. Together, these results indicate that GRA treatment can be used as an alternative to in-feed antibiotics. Further studies are needed to evaluate the viability of GRA treatment in commercial settings.

## Figures and Tables

**Table 1 animals-10-00953-t001:** Ingredient composition and nutrient contents of the nursery diets.

Ingredient and Nutrient Contents	Diets ^2^		
	Phase 1	Phase 2	Phase 3
**Ingredient, % (as fed)**			
Corn	33.75	42.91	60.65
SBM	22	29.15	34
Whey powder	27.5	17.59	-
Plasma powder	4	3.02	-
Fish meal	4.5	-	-
Fat	2	1.51	1
Dicalcium phosphate	0.4	1.51	1.4
Calcium carbonate	0.7	0.7	0.7
Salt	0.45	0.35	0.25
Swine premix ^1^	4	3	2
Zinc oxide	0.3	0.25	-
TiO_2_	0.4	0.3	0.25
**Calculated nutrient content, g/kg**			
ME, MJ/kg	14	14.1	13.7
CP (N × 6.25)	206	193	181
SID Lys ^3^	13.6	12	10.1
Ca	8.2	8.6	7.4
STTD P	5	5.2	3.8
Ca:STTD P	1.64	1.65	1.95

^1^ Providing the following amounts of vitamins and trace minerals (per kg of diet): vitamin A, 10,075 international units (IU); vitamin D3, 1100 IU; vitamin E, 83 IU; vitamin K (as menadione), 3.7 mg; D-pantothenic acid, 58.5 mg; riboflavin, 18.3 mg; choline, 2209.4 mg; folic acid, 2.2 mg; niacin, 73.1 mg; thiamin, 7.3 mg; pyridoxine, 7.3 mg; vitamin B12, 0.1 mg; D-biotin, 0.4; Cu, 12.6 mg; Fe, 100 mg; Mn, 66.8 mg; Zn, 138.4 mg; Se, 0.3 mg; I, 1.0 mg; S, 0.8 mg; Mg, 0.0622%; Na, 0.0004%; Cl, 0.0336%; Ca, 0.0634%, P, 0.003%; K, 0.0036%. ME = metabolizable energy. STTD = standardized total tract digestibility. SBM = soybean meal. ^2^ Phase 1, 2 and 3 were fed 0–7, 7–21 and 21–28 days post-weaning, respectively. ^3^ Standardized ileal digestible (SID) basis.

**Table 2 animals-10-00953-t002:** Impact of glucocorticoid receptor agonist (GRA), in-feed antibiotic (ANT) and sex on complete blood cell counts (CBC), the concentration of plasma proteins and measures of blood chemistry in pigs during the first week post-weaning.^1^

Measurement	GRA−				GRA+					*p*≤						
	ANT−		ANT+		ANT−		ANT+		SE	GRA	ANT	SEX	GRA × ANT	GRA × SEX	ANT × SEX	GRA × ANT × SEX
	Gilt	Barrow	Gilt	Barrow	Gilt	Barrow	Gilt	Barrow								
Total WBC (K/μL)	18.5	16.6	15.1	19.9	13.5	14.1	14.2	16.8	1.63	0.01	0.41	0.13	0.38	0.92	0.03	0.25
Neutrophils (K/μL)	9.4	8.7	9.5	12.6	7.7	8.7	12.1	9.3	4.20	0.7	0.16	0.91	0.86	0.51	0.99	0.23
Lymphocytes (K/μL)	6.4	6.4	4.8	5.8	5.5	4.8	5.8	5.9	0.80	0.45	0.59	0.75	0.03	0.35	0.3	0.91
N:L ratio (K/μL)	1.6	1.5	2.1	2.1	1.9	2.0	2.5	2	0.82	0.5	0.22	0.70	0.70	0.80	0.73	0.62
Monocytes (K/μL)	0.7	0.4	0.9	0.9	0.6	0.5	0.7	0.9	0.21	0.63	0.01	0.68	0.51	0.27	0.18	0.80
Eosinophils (K/μL)	1.0	0.7	0.5	0.4	0.3	0.5	0.5	0.5	0.12	0.01	0.01	0.77	0.01	0.06	0.68	0.13
RBC (M/μL)	6.8	6.8	7.1	7.1	7.2	7.1	6.8	6.9	0.24	0.69	0.91	0.73	0.07	0.93	0.62	0.46
Hemoglobin (g/dL)	10.9	10.3	10.7	10.8	10.8	10.6	9.4	10.2	0.77	0.42	0.50	0.96	0.28	0.57	0.38	0.91
Hematocrit (%)	38.7	37.4	38.8	39.1	38.7	39	34.7	37.4	3.04	0.56	0.59	0.78	0.31	0.6	0.58	0.92
Platelets (K/μL)	705.5	679.6	767.6	758.4	931.6	885.5	726.3	887.5	151.22	0.05	0.81	0.76	0.19	0.57	0.39	0.47
Plasma protein (g/dL)	6.1	5.9	5.4	5.8	5.4	4.8	6.1	5.6	0.24	0.03	0.12	0.13	0.01	0.03	0.25	0.35
Fibrinogen (mg/dL)	700.3	647.2	578.3	572.1	550.3	578.9	577	574.7	40.83	0.03	0.08	0.73	0.03	0.38	0.87	0.42
Electrolyte Balance (mEq/L)	37.9	38.5	39	38.2	39.2	38.5	38.1	38	1.25	0.95	0.66	0.69	0.26	0.86	0.66	0.32
Glucose (mg/dL)	114.9	128.8	117.7	124.9	124.6	119	111.1	130.1	9.14	0.93	0.83	0.04	0.94	0.64	0.28	0.06
BUN (mg/dL)	7.4	8.3	8.1	6.7	5.7	5.6	6.9	5.0	1.39	0.04	0.95	0.48	0.67	0.69	0.26	0.92
Creatinine (mg/dL)	1.26	1.23	1.24	1.19	1.09	1.15	1.08	1.14	0.095	0.02	0.68	0.8	0.81	0.31	0.9	0.93
Anion gap (mmol/L)	16.8	17.6	17	17.3	17.3	16.8	17.1	17.7	0.74	0.92	0.78	0.46	0.59	0.54	0.7	0.32

^1^ Data are least squares mean (± largest SE) and represent the best estimate of mean based on repeated measurements. GRA+ pigs received 0.2 mg/kg BW dexamethasone −1 and 3 days post-weaning; ANT+ pigs received in-feed tylosin (110 mg/kg diet) for the first week post-weaning. RBC = red blood cell; WBC = white blood cell; BUN = blood urea nitrogen; N:L ratio = the neutrophil-to-lymphocyte ratio.

**Table 3 animals-10-00953-t003:** Effects of glucocorticoid receptor agonist (GRA), in-feed antibiotic (ANT) and sex on intestinal mucosal morphology, the relative concentration of selected jejunum enzymes and protein, and the digestibility of dietary energy and protein in early-weaned pigs ^1^.

Measurement	GRA−				GRA+					*p*≤						
	ANT−		ANT+		ANT−		ANT+		SE	GRA	ANT	SEX	GRA × ANT	GRA × SEX	ANT × SEX	GRA × ANT × SEX
	Gilt	Barrow	Gilt	Barrow	Gilt	Barrow	Gilt	Barrow								
*Mucosal morphology*																
Villus height (mm)	0.26	0.26	0.28	0.29	0.25	0.23	0.26	0.23	0.013	0.01	0.03	0.23	0.3	0.04	0.99	0.84
Crypt depth (mm)	0.23	0.24	0.25	0.25	0.27	0.25	0.25	0.23	0.012	0.31	0.86	0.81	0.03	0.23	0.9	0.84
Villus density (villi/mm)	9.08	8.69	9.48	9.12	9.83	9.6	9.13	9.40	0.338	0.07	0.93	0.42	0.05	0.37	0.53	0.59
VH:CD	1.18	1.14	1.19	1.22	1.01	0.97	1.13	1.07	0.078	0.01	0.12	0.53	0.50	0.71	0.8	0.66
*Jejunum enzymes*																
SI (ng/mg)	117 ^a^	107 ^a,b,c^	104 ^b,c^	104 ^bc^	105 ^b,c^	113 ^a,b^	107 ^a,b,c^	103 ^b,c^	5.10	0.77	0.17	0.37	0.61	0.09	0.88	0.01
MGA (ng/mg)	6.5	6.5	7.0	7.0	8.1	8.1	7.5	7.0	1.26	0.31	0.86	0.17	0.43	0.23	0.15	0.13
APEP (pg/mg)	105	100	174	150	106	117	137	131	14.5	0.29	0.01	0.30	0.04	0.14	0.14	0.89
AP (ng/mg)	17.0	21.1	13.5	12.1	17.3	20.0	12.3	12.1	1.84	0.69	0.01	0.04	0.94	0.96	0.01	0.30
HSP-70 (ng/mg)	52.3 ^b^	52.8 ^b^	55.0 ^b^	16.9 ^b^	92.3 ^a^	94.3 ^a^	ND	ND	20.19	0.81	0.01	0.94	0.01	0.02	0.93	0.02
*Digestibility*																
ATTD of energy	0.57	0.56	0.57	0.59	0.64	0.63	0.63	0.63	0.047	0.01	0.86	0.96	0.45	0.79	0.6	0.86
AID of protein	0.74	0.77	0.84	0.89	0.93	0.89	0.77	0.60	0.059	0.7	0.11	0.49	0.01	0.11	0.34	0.22

^1^ Values reported are least squares mean (± the largest SE) and represent the best estimate of mean based on repeated measures ANOVA; ND = not detectable. ^a–c^ Means within a row lacking a common superscript letter are different (*p* < 0.05). GRA+ pigs received 0.2 mg/kg BW dexamethasone at −1 and 3 days post-weaning; ANT+ pigs received in-feed tylosin (110 mg/kg diet) for the first week post-weaning. VH:CD = villus height-to-crypt-depth ratio; SI = sucrase isomaltase; MGA = maltase glucoamylase; APEP = aminopeptidase; AP = alkaline phosphatase; HSP-70 = heat-shock protein 70; ATTD = apparent total tract digestibility; AID = apparent ileal digestibility.

**Table 4 animals-10-00953-t004:** Effects of glucocorticoid receptor agonist (GRA), in-feed antibiotic (ANT) and sex on measures of growth performance during the starter phase (28 days post-weaning).^1^

Measurement	GRA−				GRA+					*p*≤						
	ANT−		ANT+		ANT−		ANT+		SE	GRA	ANT	SEX	GRA × ANT	GRA × SEX	ANT × SEX	GRA × ANT × SEX
	Gilt	Barrow	Gilt	Barrow	Gilt	Barrow	Gilt	Barrow								
*ADG*, kg/d																
Overall	0.28	0.25	0.35	0.34	0.41	0.31	0.33	0.33	0.033	0.02	0.17	0.05	0.01	0.32	0.08	0.38
Day 7	−0.02 ^z^	0.04 ^z^	0.12 ^z^	0.12 ^z^	0.21 ^y^	0.11 ^y^	0.13 ^z^	0.06 ^y^	0.066	-	-	-	-	-	-	-
Day 14	0.24 ^y^	0.16 ^y^	0.30 ^y^	0.29 ^y^	0.33 ^y^	0.33 ^x^	0.25 ^y,z^	0.37 ^x^	0.065	-	-	-	-	-	-	-
Day 21	0.51 ^x^	0.36 ^x^	0.45 ^x^	0.45 ^x,y^	0.49 ^x^	0.41 ^x^	0.56 ^x^	0.43 ^x^	0.050	-	-	-	-	-	-	-
Day 28	0.40 ^x,y^	0.43 ^x^	0.50 ^x^	0.52 ^x^	0.62 ^x^	0.39 ^x^	0.39 ^x,y^	0.45 ^x^	0.060	-	-	-	-	-	-	-
*BW*, kg																
Overall	8.9	8.9	9.9	9.8	10.3	9.7	9.8	9.6	0.23	0.01	0.04	0.11	0.01	0.27	0.51	0.40
Day 7	7.5 ^u^	8.0 ^z^	8.5 ^u^	8.5 ^u^	9.1 ^u^	8.5 ^u^	8.6 ^u^	8.1 ^u^	0.43	-	-	-	-	-	-	-
Day 14	9.2 ^z^	9.0 ^z^	10.6 ^z^	10.5 ^z^	11.4 ^z^	10.8 ^z^	10.3 ^z^	10.6 ^z^	0.35	-	-	-	-	-	-	-
Day 21	12.8 ^y^	11.5 ^y^	13.8 ^y^	13.7 ^y^	14.8 ^y^	13.6 ^y^	14.1 ^y^	13.6 ^y^	0.44	-	-	-	-	-	-	-
Day 28	15.7 ^x^	14.5 ^x^	17.3 ^x^	17.3 ^x^	19.2 ^x^	16.3 ^x^	16.9 ^x^	16.8 ^x^	0.43	-	-	-	-	-	-	-
*ADFI*, kg/d																
Overall	0.58	0.49	0.61	0.59	0.63	0.52	0.53	0.57	0.047	0.89	0.53	0.18	0.21	0.71	0.16	0.60
Day 7	0.16 ^u^	0.18 ^u^	0.20 ^u^	0.22 ^u^	0.23 ^u^	0.18 ^u^	0.16 ^z^	0.23 ^u^	0.060	-	-	-	-	-	-	-
Day 14	0.46 ^z^	0.40 ^z^	0.44 ^z^	0.44 ^z^	0.47 ^z^	0.43 ^z^	0.33 ^y^	0.45 ^z^	0.059	-	-	-	-	-	-	-
Day 21	0.75 ^y^	0.55 ^y^	0.73 ^y^	0.72 ^y^	0.77 ^y^	0.66 ^y^	0.75 ^x^	0.70 ^y^	0.064	-	-	-	-	-	-	-
Day 28	0.96 ^x^	0.84 ^x^	1.06 ^x^	0.97 ^x^	1.03 ^x^	0.83 ^x^	0.89 ^x^	0.91 ^x^	0.063	-	-	-	-	-	-	-
*G:F*																
Overall	0.35	0.38	0.56	0.60	0.63	0.61	0.64	0.51	0.078	0.01	0.05	0.68	0.01	0.20	0.55	0.52
Day 7	−0.1 ^z^	0.19 ^z^	0.59 ^x,y^	0.71 ^x^	0.75 ^x^	0.63 ^x,y^	0.73 ^x^	0.22 ^y^	0.120	-	-	-	-	-	-	-
Day 14	0.51 ^x,y^	0.40 ^x,y,z^	0.65 ^x^	0.68 ^x^	0.72 ^x^	0.79 ^x^	0.72 ^x^	0.84 ^x^	0.150	-	-	-	-	-	-	-
Day 21	0.68 ^x^	0.64 ^x^	0.67 ^x^	0.65 ^x,y^	0.64 ^x,y^	0.64 ^x,y^	0.74 ^x^	0.62 ^x,y^	0.119	-	-	-	-	-	-	-
Day 28	0.28 ^y^	0.29 ^yz^	0.33 ^y^	0.37 ^y^	0.40 ^y^	0.38 ^y^	0.37 ^y^	0.36 ^y^	0.153	-	-	-	-	-	-	-

^1^ Overall values reported are least squares mean (± largest SE) and represent the best estimate of mean for the GRA × ANT × SEX interaction based on repeated measurement ANOVA. GRA+ pigs received 0.2 mg/kg BW dexamethasone at −1 and 3 days post-weaning; ANT+ pigs received in-feed tylosin (110 mg/kg diet) for the first week post-weaning. ^x,y,z,^^u^ Means within a column lacking a common superscript letter are different (*p* < 0.05). ADG = Average daily gain; BW = body weight; ADFI = average daily feed intake; G:F = gain-to-feed ratio.

**Table 5 animals-10-00953-t005:** Impact of glucocorticoid receptor agonist (GRA), in-feed antibiotic (ANT) and sex on measures of growth performance during the grow-to-finish phase (49 to 126 days post-weaning).^1^

Measurement	GRA−				GRA+					*p≤*						
	ANT−		ANT+		ANT−		ANT+		SE	GRA	ANT	SEX	GRA × ANT	GRA × SEX	ANT × SEX	GRA × ANT × SEX
	Gilt	Barrow	Gilt	Barrow	Gilt	Barrow	Gilt	Barrow								
*ADG*, kg/d																
Overall	0.84	0.89	0.84	0.90	0.85	0.90	0.90	0.95	0.051	0.20	0.27	0.04	0.31	0.84	0.97	0.95
Day 49	0.58 ^z^	0.60 ^z^	0.62 ^y^	0.66 ^y^	0.60 ^z^	0.60 ^z^	0.65 ^u^	0.60 ^u^	0.069	-	-	-	-	-	-	-
Day 63	0.91 ^x^	0.74 ^y,z^	0.99 ^x^	0.80 ^y^	0.83 ^y^	0.76 ^y,z^	0.83 ^z^	0.96 ^z^	0.110	-	-	-	-	-	-	-
Day 84	0.72 ^y^	0.82 ^y^	0.59 ^y^	0.76 ^y^	0.78 ^y^	0.80 ^y^	0.69 ^z,u^	0.75 ^y^	0.063	-	-	-	-	-	-	-
Day 105	0.93 ^x^	1.05 ^x^	0.88 ^x^	1.04 ^x^	1.04 ^x^	1.05 ^x^	0.98 ^y^	1.15 ^x^	0.111	-	-	-	-	-	-	-
Day 126	1.07 ^x^	1.26 ^x^	1.14 ^x^	1.22 ^x^	1.03 ^x^	1.27 ^x^	1.35 ^x^	1.27 ^x^	0.112	-	-	-	-	-	-	-
*BW*, kg																
Overall	59.5	60.6	61.6	64.1	64.4	62.2	63.0	65.1	3.33	0.01	0.04	0.33	0.23	0.3	0.1	0.41
Day 49	27.2 ^w^	28.1 ^w^	30.9 ^w^	31.8 ^w^	32.3 ^w^	29.6 ^w^	31.2 ^v^	30.0 ^v^	4.52	-	-	-	-	-	-	-
Day 63	40.2 ^u^	38.4 ^u^	44.8 ^u^	43.0 ^u^	44.0 ^u^	40.2 ^u^	42.7 ^u^	43.5 ^u^	4.20	-	-	-	-	-	-	-
Day 84	55.8 ^z^	55.5 ^z^	57.1 ^z^	58.9 ^z^	60.2 ^z^	56.9 ^z^	57.2 ^z^	59.1 ^z^	4.47	-	-	-	-	-	-	-
Day 105	75.8 ^y^	77.3 ^y^	75.7 ^y^	80.7 ^y^	82.0 ^y^	78.8 ^y^	77.6 ^y^	83.1 ^y^	4.18	-	-	-	-	-	-	-
Day 126	98.7 ^x^	103.6 ^x^	99.6 ^x^	106.2 ^x^	103.5 ^x^	105.4 ^x^	106.0 ^x^	109.7 ^x^	4.46	-	-	-	-	-	-	-
*ADFI*, kg/d																
Overall	2.33	2.46	2.3	2.55	2.45	2.44	2.35	2.56	0.059	0.3	0.56	0.01	0.81	0.21	0.02	0.45
Day 49	1.24 ^u^	1.28 ^v^	1.36 ^u^	1.36 ^v^	1.41 ^u^	1.29 ^v^	1.35 ^u^	1.38 ^u^	0.110	-	-	-	-	-	-	-
Day 63	1.78 ^z^	1.99 ^u^	1.89 ^z^	2.01 ^u^	2.09 ^z^	1.81 ^u^	1.95 ^z^	1.99 ^z^	0.120	-	-	-	-	-	-	-
Day 84	2.32 ^y^	2.46 ^z^	2.31 ^y^	2.57 ^z^	2.62 ^y^	2.42 ^z^	2.48 ^y^	2.52 ^y^	0.220	-	-	-	-	-	-	-
Day 105	2.96 ^x^	3.08 ^y^	2.71 ^y^	3.15 ^y^	2.98 ^x,y^	3.08 ^y^	2.89 ^x^	3.33 ^x^	0.210	-	-	-	-	-	-	-
Day 126	3.37 ^x^	3.5 ^x^	3.24 ^x^	3.67 ^x^	3.16 ^x^	3.58 ^x^	3.09 ^x^	3.6 ^x^	0.218	-	-	-	-	-	-	-
*G:F*																
Overall	0.38	0.38	0.38	0.37	0.36	0.38	0.4	0.38	0.032	0.676	0.391	0.669	0.244	0.635	0.237	0.447
Day 49	0.46 ^x^	0.47 ^x^	0.46 ^x^	0.49 ^x^	0.43 ^x^	0.47 ^x^	0.49 ^x^	0.43 ^x^	0.030	-	-	-	-	-	-	-
Day 63	0.50 ^x^	0.38 ^x,y^	0.52 ^x^	0.40 ^x,y^	0.40 ^x,y^	0.42 ^x,y^	0.43 ^x,y^	0.49 ^x^	0.070	-	-	-	-	-	-	-
Day 84	0.31 ^y^	0.33 ^y^	0.26 ^y^	0.29 ^y^	0.30 ^y^	0.33 ^y^	0.28 ^z^	0.30 ^y^	0.065	-	-	-	-	-	-	-
Day 105	0.31 ^y^	0.34 ^y^	0.33 ^y^	0.33 ^y^	0.35 ^x,y^	0.34 ^y^	0.34 ^y,z^	0.34 ^x,y^	0.073	-	-	-	-	-	-	-
Day 126	0.32 ^y^	0.36 ^y^	0.35 ^x,y^	0.33 ^y^	0.32 ^y^	0.36 ^y^	0.49 ^x^	0.35 ^x^	0.069	-	-	-	-	-	-	-

^1^ Overall values reported are least squares mean (± largest SE) and represent the best estimate of mean for the GRA × ANT × SEX interaction based on repeated measurement ANOVA. GRA+ pigs received 0.2 mg/kg BW dexamethasone at −1 and 3 days post-weaning; ANT+ pigs received in-feed tylosin (110 mg/kg diet) for the first week post-weaning. ^x,y,z,u,w,v^ Means within a column lacking a common superscript letter are different (*p* < 0.05). ADG = Average daily gain; BW = body weight; ADFI = average daily feed intake; G:F = gain-to-feed ratio.

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
