# Peer review of "The Effects of a Glucocorticoid Receptor Agonist (GRA) on the Immune Function, Nutrient Digestibility, and Wean-to-Finish Growth Performance of Early-Weaned Pigs"

_animals, 2020, doi:10.3390/ani10060953_

Round 1

Reviewer 1 Report

The paper describes the potential effects of dexaméthasone and tylosine on weaned pigs.

Overall, the paper is sound and interesting.

In my opinion a concern could be useless description of too complex interactions (f.e. time x main effects x sex), hard to interpret.

The authors have to discuss the effects of corticoids on body weight gain, especially water retention, that could biaise the interpretations on higher growth in DXM treated animals.

Also, they should mention the powerful effect and long half-life of DXM.

See minor comments in the attached file.

Author Response

Thank you for the comments and suggestions. We found the comments helpful and have modified the paper accordingly, as detailed below. We believe these changes result in an improved quality of the manuscript and hope they meet with your approval.

Please see our answers after >>

Reviewer 1:  The paper describes the potential effects of dexaméthasone and tylosine on weaned pigs.  Overall, the paper is sound and interesting.  In my opinion a concern could be useless description of too complex interactions (f.e. time x main effects x sex), hard to interpret.

The authors have to discuss the effects of corticoids on body weight gain, especially water retention, that could biaise the interpretations on higher growth in DXM treated animals.  Also, they should mention the powerful effect and long half-life of DXM.  See minor comments in the attached file.

>>  We have addressed the above concerns and comments of the reviewer in the specific items below.

“meaning?”  (highlighted “blocked by time” on line 77)

>> The meaning is that two groups of pigs were used in this study at different time points because of the small size of our breeding herd.  We have used the block as a fixed effect in our study to account for the differences.  Please see section 2.4 in the revised text.

“double-blind?”  (highlighted “eight treatment groups” on line 78)

>> By definition, a double-blind study is “a study in which neither the subjects [in our case pigs] nor the researchers know which treatment the subjects received until the clinical trial is over”  (NCI Medical Dictionary).  Within that definition, our study was not a double-blind study.  Although our subjects did not know what treatment they were receiving, the research technicians and staff were aware of which treatments they were distributing.  However, to avoid biases, we distributed pigs from each litter randomly among treatments.  This clarification has now been made in the text.  Please see lines ?? to ?? in the revised text.  Additionally, the pens were labeled only with numbers, thus the staff and research technicians did not know on which treatment group they were performing measurements. 

“Why?” (highlighted “lithium” for the vacutainers on line 107)

>> We used lithium-heparin, as per the instructions of the iSTAT analyzer manufacturer (Abbott Laboratories).  Please also see section 2.3.

“Why not random?” (highlighted “block” on line 178)

>> Because we included two different blocks of time within the experimental design.  Therefore, we had to account for the block as a fixed effect.  This way we could account or the fixed effect of time/block on our estimates of means.

“! Hard to interpret A X B X C (XD)” (highlighted “interaction” on line 178)

>>  We have added a statement when first mentioning the interactions that introduced our notation.  Please see lines ?? to ?? in the revised text.

“autoregressive?” (referring to lines 184-185)

>> Yes, we have used an autoregressive covariance structure.  We have added a statement to the Methods to reflect this.  Please see lines ?? to ?? in the revised text.

“Not really true” (highlighted “A strong tendency (p=0.06) for a GRAxANTxSEX effect on blood glucose concentrations was observed” on lines 225-226)

>>  The wording has been updated to indicate that a tendency was found, but not characterize it as a “strong” tendency.  Please see lines ?? to ?? in the updated text.

Standard errors on lines 234-235 were marked out.

>> We included the standard errors in these locations because they are not otherwise reported in the tables.  Thus, we thought that reporting those standard errors was critical for readers can see the degree of variability of the estimate of means. 

The unit for SI concentration was marked out on line 264.

>> We have removed the text, as suggested.  Please see lines ?? to ?? in the revised text.

“Close to trend.”  (referring to “The concentration of MGA tended to increase at day 5 post-weaning relative to days 1 and 3” on lines 273 to 275).

>> In the current study we considered 0.05 < p ≤ 0.10 as indicating a tendency or trend.  Please see section 2.4 in the Methods.  The actual p-value is 0.0998, which is below the 0.10 cutoff.  For the sake of consistency in our reporting, we rounded up to 0.10. 

“not reported” (marked through a-c Means within a row lacking a common superscript letter are different (p<0.05) in Tables 4 and 5)

>> These sentences were eliminated from the tables.  Please see the revised tables 4 and 5.

“explain” (referring to “antibiotic treatment affects the N: L ratio in yet a different way on line 411)

>>  The explanation has been provided.  We have added a connecting word to the phrase immediately after, to better communicate that the phrase after is explaining the different ways.  Please see lines ?? to ?? in the revised text.

“Need reference.  Creatinine is an indicator of MUSCLE MASS (not muscle degradation).”  (highlighted “an indicator for increased skeletal muscle degradation” on lines 448-449)

>> References have been provided showing that creatinine is the product of muscle creatine catabolism and has long been considered as an indicator of muscle protein degradation.  Please see Hosten. 1990. In Clinical methods: the history, physical, and laboratory examination. Boston, MA.  pp. 874-878; McGilvray, et.al. 2019. doi:10.1093/jas/skz120. Please also see the revised reference list.

“Speculative considering that glucocorticoids increase body water retention and modify body tissue composition.”  (bracketed text on lines 468-472)

>> It is true that one of the side-effects of long-term consumption of immunosuppressive doses of dexamethasone is increased water retention, leading to a swollen face and joints in humans.  However, in the current study, we used a low, anti-inflammatory dose (3x lower than the immunosuppressive dose for pigs – see Wooten, et. al. 2019. doi:10.1017/s1751731118003634) of dexamethasone for only two injections.  Considering that the half-life of dexamethasone is very short (plasma elimination rate of 1.06 hours in pigs – see Wyns, et. al. 2013. doi:10.1016/j.tvjl.2013.06.015), we do not think that the better performance of these pigs was due to a change in water retention.  Moreover, the superior body weight of GRA-treated pigs is maintained until the end of the starter phase, grower phase and finishing phase, long after dexamethasone would have been eliminated from the blood and tissues.  For comparison, the swollen face and joints quickly disappear in humans, once dexamethasone intake stops.

It is true that long-term glucocorticoid consumption will change body composition in favor of fat deposition.  However, at the dose and length of time used here, it would be almost impossible to increase the fat deposition in these animals.  One piece of evidence for this idea is that the G:F ratio was higher in GRA-treated pigs.  Increased fat deposition in pigs and other species is associated with a decreased G: F ratio, due to the energetic cost of fat deposition (NRC Swine 2012).  Thus, the present study provides no evidence for a change in body composition due to GRA administration.  Therefore, in the current study the higher growth performance in GRA-treated pigs could mainly be associated with a lower level of inflammation and higher levels of availability of nutrients for growth, rather than a change in body composition. 

>> We have added a discussion paragraph and other statements that addresses these ideas.  Please see the revised Discussion section.

Two uses of “in the current study” right after one another are underlined on line 467.

>> The text has been modified, as suggested.  Please see lines ?? to ?? in the revised text.

“Mucus is considered as protective against pathogens.”  (referring to text on lines 480-481)

>> Yes, mucus production is a part of the innate immune system.  However, inflammation will increase mucus production, which is the main component of endogenous losses, negatively impacting the availability of proteins and amino acids for growth and production.  Also, early weaning-induced inflammation causes damage to the intestinal lining.  What has been stated in the discussion section does not imply that a reduction in mucus production in GRA-treated pigs reflects a limitation in immunoprotective measures.  It implies that the anti-inflammatory effects reduced damage to the intestine and the production of mucus and thus reduced endogenous protein losses.  The latter suggests more available protein and amino acids for growth in GRA-treated pigs.  We have revised our statement in the discussion to reflect this idea.  Please see lines ?? to ?? in the revised text.

“Water retention?”  (referring to discussion on lines 500 – 502)

>>  Please see our answer above to the other water retention comment made by this reviewer.

“Discuss half-life of dexamethasone”  (written next to the last paragraph of the discussion)

>>  A discussion of the implications of the dexamethasone half-life has now been added.  Please see the revised Discussion.

Reviewer 2 Report

Please explain why tylosin was used for comparison, when it is not allowed to be used for growth promotion or feed efficiency in the US. Probiotics are commonly added to starter feeds for these purposes.

Experimental unit for GRA is the pig, but experimental unit for ANT is the pen, since pigs within a pen cannot be randomly assigned to different ANT treatments. The experiment should thus be analyzed as a split plot. This was not clear in the methods.

Include dose of tylosin in the methods.

Author Response

Thank you for the comments and suggestions. We found the comments helpful and have modified the paper accordingly, as detailed below. We believe these changes result in an improved quality of the manuscript and hope they meet with your approval.

Please see our answers after >>

Reviewer 2:  Please explain why tylosin was used for comparison, when it is not allowed to be used for growth promotion or feed efficiency in the US. Probiotics are commonly added to starter feeds for these purposes.

Experimental unit for GRA is the pig, but experimental unit for ANT is the pen, since pigs within a pen cannot be randomly assigned to different ANT treatments. The experiment should thus be analyzed as a split plot. This was not clear in the methods.  Include dose of tylosin in the methods.

>> We have addressed the above concerns and comments of the reviewer in the specific items below.

Grammar mistakes were highlighted on the following lines: 20-21, 36, 64, 67, 76

>> All of the grammar mistakes above have been corrected.  See the revised text.  We believe that grammar highlighted on lines 38-39 and 186 is correct.  However, we’re more than happy to work with animals copy editors to ensure that this is case.

“I thought a pig can only be slaughtered once!” (highlighted on line 34)

>>  Our use of “serial slaughter” is referring to the serial slaughter technique, a common terminology that refers to slaughtering a small number of animals within a treatment group at different time points (NRC Swine 2012).  The text has been updated to better reflect our methodology.  Please see lines ?? to ?? in the revised text.

 “subsiding” (highlighted on line 44).

>> We have changed the word to reduced.

“Using parallel structure would make the results much clearer.” (comment on line 40)

>> We have adjusted the wording to give parallel structure.  Please see lines ?? to ?? in the revised text.

“Data collection is only mentioned during the first week.” (comment on line 42)

>> The measures of growth performance referred to on line 42 were collected throughout the study.  Specifically, body weight and feed intake were measured daily from days 1 – 7, weekly from days 7 – 28, and every three weeks from days 28 – 126, as described in Section 2.1 of the Methods.  Please see lines ?? to ?? of the revised text.

“This use is no longer allowed in the US.” (comment on line 56)

>> Although the reviewer is correct that the USDA no longer allows antibiotic use as a growth-promoter in pigs, producers still regularly use antibiotics to help piglets cope with the immunological stress of weaning.  Indeed, this study was done in response to recent calls from the USDA for research into alternatives to antibiotic use.  However, our use of the word “sub-therapeutic” is likely confusing in this context and also incorrect.  Therefore, we have removed the word “sub-therapeutic” throughout the manuscript.  Please see the revised text.

>> Tylosin was used in this study because it is a wide-spectrum antibiotic that is widely used in the U.S. to mitigate the negative effects of the stress of weaning in pigs. 

“Probiotics are commonly added to starter feed for this.” (comment on line 59)

>> We agree with the reviewer that probiotics are being used an alternative to antibiotics in some pig operation units.  The effects of probiotics and prebiotics on the health and productivity of pigs has been extensively studied (NRC Swine 2012).  However, the effects of probiotics or prebiotics has been shown to be dependent upon the management conditions of the pig operation.  Thus, it’s not always effective at mitigating the stress of weaning in piglets (NRC Swine 2012; Liao & Nyachoti. 2017. doi:10.1016/j.aninu.2017.06.007; Zimmermann, et. al. 2016. doi:10.1016/j.anifeedsci.2016.06.021).  Therefore, the objective of this study was to compare the viability of GRA to antibiotics.  This comparison was particularly relevant since antibiotics cause food safety concerns, but to our knowledge, probiotics do not.  However, we agree that future research could compare the effects of GRA-treatment to probiotics, prebiotics, anthelmintics, organic acids and other feed additives.   

“certain” (highlighted on line 61).

>> We have changed the word to specific.

“Have they been compared to probiotics in the feed?” (comment on line 63)

>> Not to our knowledge.  However, it’s a very good research topic for the future.  We may explore that comparison in our laboratory in the future.

“With appropriate veterinary prescription [for the antibiotic], I hope.” (comment on line 74)

>> Yes.  As stated in the same paragraph, the experimental protocol was approved by the TTU Animal Care and Use Committee of Texas Tech University.  This committee ensures that all relevant laws and regulations are followed before approving any research protocols.  In addition, two veterinarians are a part of this committee and provide prescriptions for items, such as antibiotics, when needed.

“Dose?  Requires VFD prescription.” (comment on line 82)

>> The dose of tylosin has been added.  See lines ?? to ?? in the revised text.

>> As stated above, all tylosin use was done with a veterinary prescription and with approval from the Texas Tech University Animal Care and Use Committee.

“This makes pen the experimental unit for in-feed tylosin treatment” (comment on line 85)

>> As mentioned in section 2.4, we did use pen as the experimental unit for measures of growth performance.  But, for all other measurements for individual animals, pig was used as the experimental unit.  It would be inaccurate and inappropriate to pool measurements, such as those for blood samples, for individual each pen and then use pen as the experimental unit, since differences in the animals are controlled by more variables (such as genetic factors) than our treatments.  To pool measurements together and use pen as the experimental unit would lead to an inaccurate estimation of means.  Thus, the more appropriate model would be using pig as the experimental unit consistently among the different treatment groups for measurements such as CBCs, blood chemistry, enzyme analysis, histomorphology, etc.  (Please see Biostatistics for animal science: an introductory text.  M. Kaps & W. Lamberson.  CAB International. Oxfordshire, UK. pp. 382-393)

“CP and ME should be measured, because they can vary in the ingredients.” (comment in Table 1)

>> We did a proximate analysis of the diets and all the measurements were within 5% of the calculated values.  We have added a statement to the results section stating this.

“Not correct” (highlighted 0.195 in the Ca:STTD P column)

>> Done.  Please see the corrected version of Table 1 in the revised document.

“At random times?” (referring to the collection of serial blood samples on line 104)

>> Serial blood samples were collected every other day for the first-week post-weaning, beginning at day 0 post-weaning.  The text has been updated to include these specifics.  Please see lines ?? to ?? in the revised text.

“What was done with the outliers?” (comment on line 177)

>> Typically in our laboratory, data from outliers are excluded from analysis as a standard procedure.  However, we found no outliers to be excluded in this study.  A sentence has been added to this section to better reflect this.  Please see lines ?? to ?? in the revised text.

“Should be analyzed as split-plot since the pig is the experimental unit for GRA, but pen is the experimental unit for ANT.” (comment on line 178)

>> We respectfully disagree with analyzing our data in a split-plot design because it leads to an inaccurate estimation of means (Please see our previous response to the reviewer’s comment on line 85, as well as Biostatistics for animal science: an introductory text.  M. Kaps & W. Lamberson.  CAB International. Oxfordshire, UK. pp. 382-393) However, to better clarify our methods, we need to mention that we did account for the effects of the pens by incorporating pig within pen (pig X pen) as a random effect.  We have a statement clarifying this point.  Please see lines ?? to ?? in the revised document.

“blood chemistry” (highlighted on line 247)

>>  No comments were given for this highlighted item; therefore, we were unclear as to what changes were needed.  However, we did check the text and the data in Table S2 and all appears to be correct.

“What was the limit of detection [of Hsp-70 levels]?” (comment on line 311)

>> The detection limit, per the manufacture’s label, was 5 – 100 ng/ml.  The values that were undetectable ranged between 0 to 50 ng/ml.  However, when the average of the technical replications was tested against zero, the difference was not statistically significant.  Therefore, they were considered non-detectable.

Several p-values are highlighted in Table 4.

>> No comments were given for these highlighted items; therefore, we were unclear as to what changes were needed.  However, we did check the data in Table 4 and all appears to be correct.  We have reported any possible main effects or two-way interactions in the text.

“Reference?” (comment on line 385)

>> Relevant citations have been provided.  Please see lines ?? to ?? in the revised text.

“7 days is not long-term sub-therapeutic use” (comment on line 392)

>>  We agree with the reviewer and have altered our wording accordingly.  Please see lines ?? to ?? in the revised text.

“Which are not allowed in the US.  Further studies are needed to compare GRA to in-feed probiotics. (comment on line 527)

>> As mentioned previously, although the reviewer is correct that the USDA no longer allows antibiotic use as a growth-promoter in pigs, producers still regularly use antibiotics to help piglets cope with the immunological stress of weaning.  Indeed, this study was done in response to recent calls from the USDA for research into alternatives to antibiotic use.  However, our use of the word “sub-therapeutic” is likely confusing in this context and also incorrect.  Therefore, we have removed the word “sub-therapeutic” throughout the manuscript.  Please see the revised text.

>> It would be ideal to compare GRA treatment to other potential alternatives to antibiotics, including probiotics.  However, the objective of this study was to compare the viability of GRA to antibiotics and a comparison between GRA and other potential feed additives for newly weaned pigs was beyond the scope of this current study.  However, we agree with the reviewer that further studies that would compare GRA treatment to other feed additives, including probiotics, are needed when exploring the viability of GRA treatment.

Round 2

Reviewer 2 Report

I disagree with the authors' statement that "But, for all other measurements for individual animals, pig was used as the experimental unit. It would be inaccurate and inappropriate to pool measurements, such as those for blood samples, for individual each pen and then use pen as the experimental unit, since differences in the animals are controlled by more variables (such as genetic factors) than our treatments." This confuses the measurement unit or individual animal, with the experimental unit, which is the unit that is randomly assigned to treatment, in the case the pen for GRA. Differences in the animals are appropriately controlled by randomly assigning individuals to treatments, but this was not done within pen for in-feed tylosin. In addition, if GRA is not effective for pens of pigs, it is not appropriate for commercial production.

Since what was done is clear, I am willing to let readers form their own opinions.
